

# Phylogenetic position of the enigmatic genus *Atherospio* and description of *Atherospio aestuarii* sp. nov. (Annelida: Spionidae) from Japan

Hirokazu Abe[1,2] and Kotaro Kan[1]

[1] Department of Biology, Center for Liberal Arts & Sciences, Iwate Medical University, Yahaba-cho, Shiwa-gun, Iwate, Japan
[2] Faculty of Science and Engineering, Ishinomaki Senshu University, Ishinomaki, Miyagi, Japan

## ABSTRACT

**Background:** There are currently two species within the small enigmatic genus *Atherospio* Mackie & Duff, 1986, which belongs to the *Pygospiopsis-Atherospio* group in the family Spionidae Grube, 1850. The taxonomic relationship of the genus *Atherospio* with other spionid or spioniform genera is currently not well understood due to its unusual morphological characteristics.

**Methods:** Here, we describe a new *Atherospio* species, *Atherospio aestuarii* **sp. nov.**, based on materials collected from three localities in Japan: Hirota Bay (Iwate Prefecture), Ago Bay (Mie Prefecture), and Yakushima Island (Kagoshima Prefecture). We have also evaluated the possible systematic position of this new species by conducting molecular phylogenetic analyses using the nuclear 18S, 28S, and mitochondrial 16S rRNA gene sequences.

**Results:** The morphology of *A. aestuarii* **sp. nov.** resembles that of *A. disticha* Mackie & Duff, 1986 and *A. guillei* (Laubier & Ramos, 1974) in having branchiae fused to the notopodial lamellae on a restricted number of segments from chaetiger 7, modified neurochaetae on chaetiger 5, and at least some bidentate neuropodial hooks with the secondary tooth below the main fang. The form and arrangement of the modified aristate neurochaetae in double vertical rows closely resemble those found on chaetigers 4 and 5 of *A. disticha*. The new species lacks the occipital antenna present in *A. disticha*. In this respect it resembles *A. guillei*, however, that species differs in having robust neuropodial spines on chaetiger 5 and peristomial papillae, and a preponderance of unidentate neurochaetae. Both *A. guillei* and the new species have slender needle-like notochaetae in their posteriormost chaetigers. *Atherospio aestuarii* **sp. nov.** is distinguished from both congeneric species by its branchial and neuropodial hook distributions. The new species is also unique in that it was recorded at relatively shallow depths, which included intertidal zones. The results of our molecular phylogenetic analysis indicate that the new species was included in a clade that included the genera of the *Polydora* complex, *Pygospio* Claparède, 1863, *Glandulospio* Meißner, Bick, Guggolz, Götting, 2014, *Spio* Fabricius, 1785, *Microspio* Mesnil, 1896, *Marenzelleria* Mesnil, 1896, *Rhynchospio* Hartman, 1936, *Scolelepis* Blainville, 1828, *Dispio* Hartman, 1951, and *Malacoceros* Quatrefages, 1843 with robust statistical support. The new species formed a clade with *Dispio* and *Scolelepis*, however, statistical support for the node was not significant.

Corresponding author
Hirokazu Abe, habe@isenshu-u.ac.jp

## INTRODUCTION

*Atherospio Mackie & Duff, 1986* is a small genus in the family Spionidae Grube, 1850 that currently consists of two species: *A. disticha Mackie & Duff, 1986* and *A. guillei* (*Laubier & Ramos, 1974*). The genus is closely related to *Pygospiopsis Blake, 1983* (including the recently synonymized genus *Pseudatherospio* Lovell, 1994: *Blake & Maciolek, 2018*) as it has similar prostomial shapes, an occipital antenna, modified anterior neurochaetae, branchiae that are either basally or entirely fused to the notopodial lamellae, and unusual bidentate neuropodial hooks. In other spionids, the small tooth (teeth) of the neuropodial hooded hooks is (are) superior to the main fang on the convex side, while for *Atherospio* and *Pygospiopsis* the neuropodial hooded or unhooded hooks have a small tooth or knob on the concave side, which is subapical to the terminal shaft or main fang (*Blake & Maciolek, 2018*). *Atherospio*, *Pygospiopsis*, and a recently established genus, *Aciculaspio Blake & Ramey-Balci, 2020*, are collectively called the *Pygospiopsis-Atherospio* group (*Blake & Ramey-Balci, 2020*) and currently consist of nine species. *Atherospio* and *Pygospiopsis* are distinguishable as the former have their first branchiae on chaetiger 7, while the latter simple or partially fused branchiae anterior to chaetiger 7 in a variety of patterns (*Blake & Maciolek, 2018*). *Aciculaspio* differs from both *Atherospio* and *Pygospiopsis* as it has enlarged flattened branchiae fused to the dorsal lamellae from chaetiger 10 instead of 7 and simple, unidentate-hooded hooks with curved and pointed fangs (*Blake & Ramey-Balci, 2020*).

*Blake, Maciolek & Meißner (2020)* divided the spionid genera into four clades following *Blake & Arnofsky (1999)* and *Blake (2006)*: (1) Subfamily Nerininae Söderström, 1920; (2) Subfamily Spioninae Söderström, 1920; (3) Clade consisting of *Pygospiopsis*, *Atherospio*, and *Pseudatherospio* (= *Pygospiopsis-Atherospio* group); and (4) five monotypic genera with no strong affinity for other spionids (*Glandulospio Meißner et al., 2014*; *Glyphochaeta* Bick, 2005; *Spiogalea* Aguirrezabalaga & Ceberio, 2005; *Spiophanella* Fauchald & Hancock, 1981; and *Xandaros* Maciolek, 1981). Species belonging to the *Pygospiopsis-Atherospio* group are superficially similar to species in subfamily Spioninae (including the *Polydora* complex and the genera *Pygospio* Claparède, 1863, *Microspio* Mesnil, 1896, and *Spio* Fabricius, 1785), some of which were originally classified as separate genera within Spioninae. *Pygospiopsis dubia* (Monro, 1930) was originally described as *Pygospio*, and *Blake (1983)* later established the genus *Pygospiopsis* for this species. *Atherospio guillei* was originally described as *Polydora* Bosc, 1802 in the *Polydora* complex, and later, *Meißner & Bick (2005)* transferred this species to *Atherospio*. *Atherospio guillei* and the species belonging to the *Polydora* complex both have heavy spines in the fifth segment. However, this is not considered to be evidence of a close relationship between the two taxa as these heavy spines are not homologous *sensu* stricto,

as in *A. guillei* they are neuropodial, while in *Polydora* complex they are notopodial (*Mackie & Duff, 1986*; *Radashevsky & Fauchald, 2000*; *Radashevsky, 2012*).

The close relationship between the *Pygospiopsis-Atherospio* group and the subfamilies Spioninae and Nerininae has not been consistently supported in previous studies. The first phylogenetic analysis of the Spionidae genera using morphology by *Sigvaldadóttir, Mackie & Pleijel (1997)* indicated that there were four clades in the family: (1) *Aonidella* López-Jamar, 1989 and *Xandaros*; (2) *Prionospio* complex, *Laonice* Malmgren, 1867, *Spiophanes* Grube, 1860, and *Aonides* Claparède, 1864; (3) a large unresolved assemblage of genera including the *Polydora* complex, *Scolelepis* Blainville, 1828, *Malacoceros* Quatrefages, 1843, and *Spio*; and (4) *Atherospio*, *Pseudatherospio*, and *Pygospiopsis*, but the support for these clades was weak and the selection of outgroups was subsequently deemed unfortunate (*Blake, Maciolek & Meißner, 2020*). *Mackie (1996)* re-examined the intergeneric relationships within Spionidae examined by *Sigvaldadóttir, Mackie & Pleijel (1997)* by adding several new taxa and eight taxa with questionable generic attribution using the same outgroups and indicated generally consistency with the previous results, but separated the third large unresolved assemblage by *Sigvaldadóttir, Mackie & Pleijel (1997)* into a 'polydorid' group (including *Pygospio muscularis* Ward, 1981 and excluding *Tripolydora* Woodwick, 1964 and *Pseudopolydora primigenia Blake, 1983*) and a large group including 11 genera. He also showed that the *Pygospiopsis-Atherospio* group, which includes two unnamed groups provisionally assigned as 'Genus A' and 'Genus B', is sister to a clade consisting of *Pseudopolydora primigenia* and *Pygospio elegans* Claparède, 1863. The third phylogenetic analysis of the spionid genera using morphological, reproductive, and developmental characteristics from *Blake & Arnofsky (1999)* indicated that there were three clades: two major clades consisting of the subfamily Spioninae and a larger clade consisting of all remaining spionid genera and the genera *Heterospio* Ehlers, 1874 (now considered to be a taxon closely related to cirratuliform polychaetes rather than spioniforms: *Blake & Maciolek, 2019*), *Poecilochaetus* Claparède in Ehlers, 1875, *Trochochaeta* Levinsen, 1884, and *Uncispio* Green, 1982, and a third, minor clade consisting of the enigmatic genus *Pygospiopsis* (including *Atherospio*). At present, because of several unusual morphological characteristics of the *Pygospiopsis-Atherospio* group, its taxonomic relationship with other spionids or spioniforms is not well understood. However, *Blake & Maciolek (2018)* noted that the large recurved hooded hooks of *P. profunda Blake & Maciolek, 2018* have similarities with the giant modified neuropodial hooks or spines of some *Uncispio* species.

To date, there are no available molecular data on the *Pygospiopsis-Atherospio* group or *Uncispio* deposited in the DNA Data Bank of Japan (DDBJ), the European Nucleotide Archive (ENA), or GenBank databases. Therefore, these taxa were not included in the first and recent comprehensive molecular phylogenetic analyses of the spionid genera, which was conducted by *Abe & Sato-Okoshi (2021)* and *Wang et al. (2022)*, respectively. Our field surveys have identified several specimens of the genus *Atherospio*, which have never been recorded from Japan before, from several study sites. In this study, we report the morphology of the specimens and compare it with that of other species of the genus, and describe a new species, *Atherospio aestuarii* **sp. nov.** We also evaluate the

phylogenetic position of *Atherospio* by conducting the first molecular phylogenetic analysis including the genus, whose phylogenetic position has remained a question until now.

## MATERIALS AND METHODS

### Specimen collection

Specimens of the *Atherospio* species were collected from bottom sediments in the intertidal zone of Otomo-ura (38.9958 N, 141.6817 E), Hirota Bay, Iwate Prefecture on August 6, 2017, August 18, 2018, and August 4, 2020; subtidal zones <1 m in depth in a nameless small inlet of Ago Bay (34.2985 N, 136.8311 E), Mie Prefecture on October 8, 2021; and a small fishing port at the mouth of the Kurio River (30.2741 N, 130.4214 E) on Yakushima Island, Kagoshima Prefecture, Japan on November 6, 2021 (Figs. 1 and 2). The water areas where the specimens were collected in this study are not protected, and no permission of any kind is required to collect the organisms. In the field survey of this study, we did not collect any commercially marine species and did not use any collection method that violated the prefectural fishery regulation, so we did not need any permission for the survey.

### Morphological observation

Specimens were observed and photographed in a live condition and then fixed in 10% neutral formalin seawater or 70% ethanol for morphological and molecular analyses. The morphology of the living and fixed *Atherospio* species was observed under a stereomicroscope (LW-820T; Wraymer, Osaka, Japan) and phase-contrast microscope (Eclipse 80i; Nikon, Tokyo, Japan). Light micrographs were obtained using a digital camera (Sony α6000, Tokyo, Japan) attached to the microscope. Live specimens were anesthetized in a 7% magnesium chloride solution if required. Four specimens were stained with a solution of methyl green in ethanol for light microscopy analysis. The type materials were deposited in the National Museum of Nature and Science (NSMT), Tsukuba, Japan, under the following museum registration numbers: NSMT-Pol H-858 and P-859–866.

The electronic version of this article in Portable Document Format (PDF) will represent a published work according to the International Commission on Zoological Nomenclature (ICZN), and hence the new names contained in the electronic version are effectively published under that Code from the electronic edition alone. This published work and the nomenclatural acts it contains have been registered in ZooBank, the online registration system for the ICZN. The ZooBank LSIDs (Life Science Identifiers) can be resolved and the associated information viewed through any standard web browser by appending the LSID to the prefix http://zoobank.org/. The LSID for this publication is: urn:lsid:zoobank.org:pub:ED1D54BF-7C4E-4277-A675-F604C743E6C7. The online version of this work is archived and available from the following digital repositories: PeerJ, PubMed Central SCIE and CLOCKSS.
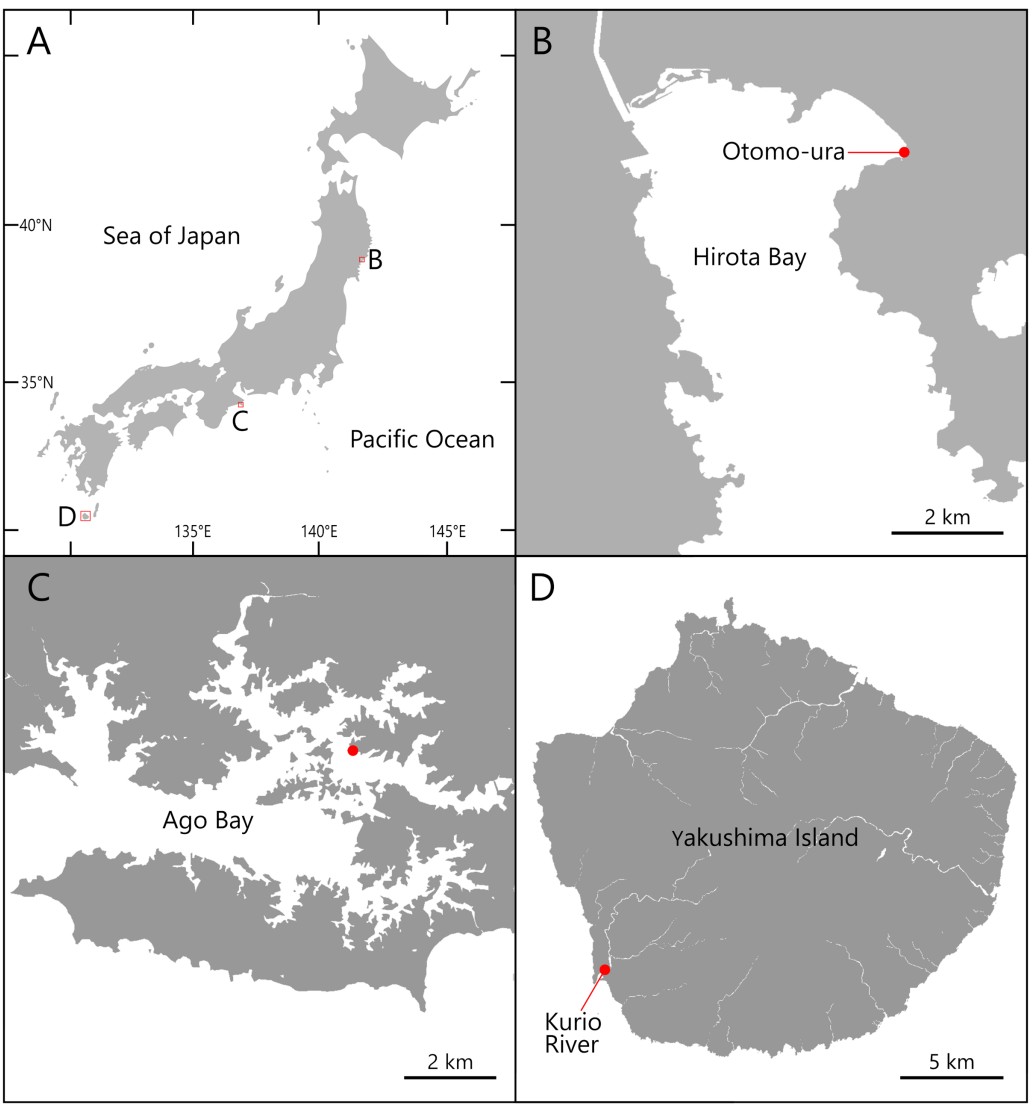

**Figure 1** **Maps of the sampling localities of *Atherospio aestuarii* sp. nov.** (A) Japan. (B) Hirota Bay. (C) Ago Bay. (D) Yakushima Island.

## Molecular analysis

Nuclear 18S, 28S, and mitochondrial 16S rRNA gene analyses were performed on the holotype and the six paratypes. Genomic DNA was extracted from 70% ethanol-preserved tissue by grinding and heating at 95 °C for 20 min in 50 µL TE buffer (pH 8.0) with 10% Chelex 100 (Bio-Rad, Hercules, CA, USA), according to *Richlen & Barber (2005)*. Ten-fold diluted extracted DNA in TE buffer was used as a template for polymerase chain reaction (PCR). Partial sequences of the nuclear 18S, 28S, and mitochondrial 16S rRNA genes were amplified by PCR using the primer pairs 18S-1F1/18S-1R632, 18S-2F576/18S-2R1209, and 18S-3F1129/18S-R1772 for 18S (*Nishitani et al., 2012*), D1R/D2C for 28S (*Scholin et al., 1994*), and 16Sar/16Sbr for 16S (*Palumbi et al., 1991*). PCR was performed in a 10 µL reaction mixture containing 0.5 µL of template DNA, 4 µL of sterilized water, 5 µL of

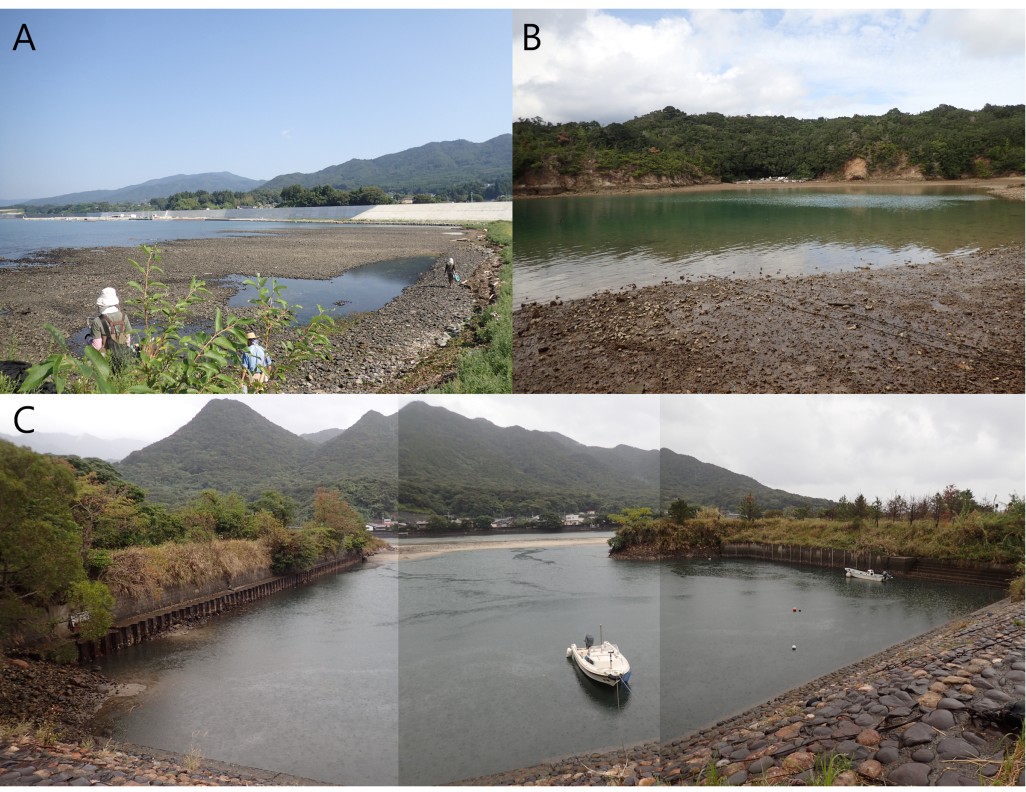

**Figure 2** **Photos of the sampling localities of *Atherospio aestuarii* sp. nov.** (A) Otomo-ura in Hirota Bay, Iwate Prefecture. (B) A nameless small inlet in Ago Bay, Mie Prefecture. (C) A small fishing port at the mouth of the Kurio River in Yakushima Island, Kagoshima Prefecture.

2 × KOD One PCR Master Mix (TOYOBO, Osaka, Japan), and 0.05 µM of 50 µM forward and reverse primers. The PCR cycling conditions were 36–40 cycles of denaturation at 98 °C for 10 s, annealing at 54 °C or 56 °C (16S), or 60 °C (18S and 28S) for 5 s, and extension at 68 °C for 1 s. PCR products were purified using Enz-Sap (Edge BioSystems, San Jose, CA, USA) and sequenced by Eurofins Genomics (Tokyo, Japan). Forward and reverse complementary sequences and contigs were assembled using GeneStudio ver. 2.2.0.0 (GeneStudio, Inc. Suwanee, GA, USA). All sequences generated in this study have been deposited in the DDBJ/ENA/GenBank nucleotide sequence database under accession numbers LC685029–LC685049 (Table 1).

To reconstruct the molecular phylogeny, sequences of the 18S, 28S, and 16S rRNA genes were aligned with the sequences of other spionid species and outgroups obtained from GenBank (Table 1) using the MAFFT online service ver. 7 (*Katoh, Rozewicki & Yamada, 2017*) and the L-INS-i algorithm. The gene sequences of the sabellid species *Amphicorina mobilis* (Rouse, 1990) and *Sabella pavonina* Savigny, 1822, obtained from DDBJ/ENA/GenBank, were used as the outgroup taxa (Table 1). Ambiguously aligned regions were eliminated using the Gblocks server ver. 0.91b with the least stringent settings (*Castresana, 2000*; *Talavera & Castresana, 2007*). The final lengths of the aligned sequences were 1,703, 663, and 434 bp for the 18S, 28S, and 16S rRNA gene sequences,

Table 1 **Terminal taxa of spionid species and outgroups (Sabellidae) used in the phylogenetic analyses and the DDBJ/EMBL/GenBank accession numbers, together with the museum registration number of the specimens used in the present study.** The organism names of unidentified species are labeled with the identifiers in the DDBJ/EMBL/GenBank database. The classifications defined by *Blake, Maciolek & Meißner (2020)* and *Wang et al. (2022)* are also provided. The gene sequences obtained in this study are highlighted in boldface type.

| Classification by *Blake, Maciolek & Meißner (2020)* | Classification by *Wang et al. (2022)* | Genus | Species | Locality | Museum registration number | Accession number | | | Reference |
|---|---|---|---|---|---|---|---|---|---|
| | | | | | | 18S | 28S | 16S | |
| *Pygospiopsis-Atherospio* Group | – | *Atherospio* | *Atherospio aestuarii* **sp. nov.** | Japan (Otomo-ura) | NSMT-Pol P-861 | **LC685029** | **LC685036** | **LC685043** | This study |
| | | | | Japan (Ago Bay) | NSMT-Pol P-862 | **LC685030** | **LC685037** | **LC685044** | This study |
| | | | | Japan (Ago Bay) | NSMT-Pol P-863 | **LC685031** | **LC685038** | **LC685045** | This study |
| | | | | Japan (Ago Bay) | NSMT-Pol P-864 | **LC685032** | **LC685039** | **LC685046** | This study |
| | | | | Japan (Kurio River) | NSMT-Pol H-858 | **LC685033** | **LC685040** | **LC685047** | This study |
| | | | | Japan (Kurio River) | NSMT-Pol P-865 | **LC685034** | **LC685041** | **LC685048** | This study |
| | | | | Japan (Kurio River) | NSMT-Pol P-866 | **LC685035** | **LC685042** | **LC685049** | This study |
| Subfamily Nerininae | Subfamily Nerininae | *Aonidella* | *Aonidella* cf. *dayi* Maciolek in López-Jamar, 1989 | NE Atlantic | | KF434504 | – | KF434508 | *Meißner et al. (2014)* |
| | | *Aonides* | *Aonides oxycephala* (Sars, 1862) | France | | MG913226 | MG878926 | MG878895 | V. Radashevsky et al. (2018, unpublished data) |
| | | *Aurospio* | *Aurospio dibranchiata* Maciolek, 198 | Kaplan, Pacific Mn nodule province | | EU340091 | – | EU340087 | *Mincks et al. (2009)* |
| | | | *Aurospio foodbancsia Mincks et al. (2009)* | West Antarctic Peninsula shelf | | EU340097 | – | EU340078 | *Mincks et al. (2009)* |
| | | *Laonice* | *Laonice* sp. VR-2006 | Sweden | | DQ779655 | DQ779693 | DQ779619 | *Rousset et al. (2007)* |
| | | *Paraprionospio* | *Paraprionospio coora* Wilson, 1990 | Japan | | LC545859 | – | LC595689 | *Abe & Sato-Okoshi (2021)* |
| | | | *Paraprionospio patiens* Yokoyama, 2007 | Japan | | LC545861 | – | LC595691 | *Abe & Sato-Okoshi (2021)* |
| | | *Poecilochaetus* | *Poecilochaetus serpens* Allen, 1904 | France | | AY569652 | – | AY569680 | *Bleidorn, Vogt & Bartolomaeus (2005),* |
| | | | *Poecilochaetus* sp. VR-2006 | France | | DQ779667 | DQ779705 | DQ779630 | *Rousset et al. (2007)* |
| | | *Prionospio* | *Prionospio dubia* Day, 1961 | USA | | EU418859 | EU418867 | – | *Struck et al. (2008)* |
| | | | *Prionospio* sp. C *sensu* Guggolz et al. (2020) (as *Prionospio* sp. 29 PB) | Clarion–Clipperton Fracture Zone | | MK971148 | – | MK971035 | *Bonifácio, Martínez-Arbizu & Menot (2020)* |
| | | | *Prionospio* sp. E *sensu* Guggolz et al. (2020) (as *Prionospio ehlersi*) | CROZEX | | EU340095 | – | EU340081 | *Mincks et al. (2009)* |
| | | | *Prionospio* sp. KJO-2005 | USA | | DQ209226 | DQ209246 | – | *Osborn et al. (2007)* |
| | | *Spiophanes* | *Spiophanes* cf. *convexus* Delgado-Blas, Díaz-Díaz & Viéitez, 2019 | France | | MG913229 | MG878931 | MG878902 | *Radashevsky et al. (2020a)* |

(Continued)

| Classification by *Blake, Maciolek & Meißner (2020)* | Classification by *Wang et al. (2022)* | Genus | Species | Locality | Museum registration number | Accession number | | | Reference |
|---|---|---|---|---|---|---|---|---|---|
| | | | | | | **18S** | **28S** | **16S** | |
| | | | *Spiophanes uschakowi* Zachs, 1933 | Russia | | KM998760 | MG878949 | MG878915 | *Radashevsky et al. (2020a)* |
| | | *Streblospio* | *Streblospio* sp. | India | | KY704336 | KY704324 | KY704328 | T. Vijapure et al. (2017, unpublished data) |
| | | *Trochochaeta* | *Trochochaeta multisetosa* (Örsted, 1844) | Norway | | MN296517 | – | MN193552 | *Radashevsky et al. (2020a)* |
| | | | *Trochochaeta* sp. THS-2006 | – | | DQ790097 | DQ790070 | – | *Struck et al. (2007)* |
| | Subfamily Spioninae | *Dispio* | *Dispio remanei* Friedrich, 1956 | Brazil | | KU900474 | KU900467 | – | M. Rebelo & M. Schettini (2016, unpublished data) |
| | | *Malacoceros* | *Malacoceros fuliginosus* (Claparède, 1868) | France/Germany | | AY525632 | – | EF431961 | *Struck & Purschke (2005)*, *Blank & Bastrop (2009)* |
| | | | *Malacoceros* cf. *indicus* (Fauvel, 1928) | Japan | | LC545857 | – | LC595687 | *Abe & Sato-Okoshi (2021)* |
| | | | *Malacoceros* sp. V040 | Germany | | MN215953 | MN215954 | – | *Surugiu, Schwentner & Meißner (2022)* |
| | | *Marenzelleria* | *Marenzelleria arctia* (Chamberlin, 1920) | Russia | | KJ546264 | KJ546214 | KJ546306 | *Radashevsky et al. (2014)* |
| | | | *Marenzelleria viridis* (Verrill, 1873) | USA/Danmark | | EU418860 | EU418868 | DQ309252 | *Struck et al. (2008)*, *Bastrop & Blank (2006)* |
| | | *Rhynchospio* | *Rhynchospio arenicola* Hartman, 1936 | USA | | KJ546286 | KJ546236 | KJ546318 | *Radashevsky et al. (2014)* |
| | | | *Rhynchospio* cf. *foliosa* Imajima, 1991 (as *Rhynchospio foliosa*) | USA | | KP986489 | KP986490 | KP986488 | *Radashevsky et al. (2016a)* |
| | | *Scolelepis* | *Scolelepis squamata* (Müller, 1806) | Spain | | MN215944 | MN215960 | – | *Surugiu, Schwentner & Meißner (2022)* |
| | | | *Scolelepis texana* Foster, 1971 | Japan | | LC545882 | – | LC595712 | *Abe & Sato-Okoshi (2021)* |
| Incertae sedis | | *Glandulospio* | *Glandulospio orestes* Meißner et al., 2014 | NE Atlantic | | KF434505 | – | KF434511 | *Meißner et al. (2014)* |
| Subfamily Spioninae | | *Boccardia* | *Boccardia proboscidea* Hartman, 1940 | Japan | | LC107607 | AB973944 | LC595721 | *Abe, Kondoh & Sato-Okoshi (2016)*, *Simon, Sato-Okoshi & Abe (2019)*, *Abe & Sato-Okoshi (2021)* |
| | | | *Boccardia pseudonatrix* Day, 1961 | France | | LC682681 | LC682702 | LC682725 | W. Sato-Okoshi et al. (2022, unpublished data) |
| | | *Boccardiella* | *Boccardiella hamata* (Webster, 1879) | France | | LC682684 | LC682705 | LC682727 | W. Sato-Okoshi et al. (2022, unpublished data) |
| | | *Dipolydora* | *Dipolydora bidentata* (Zachs, 1933) | Russia | | JX228065 | JX228085 | JX228103 | *Radashevsky & Pankova (2013)* |
| | | | *Dipolydora giardi* (Mesnil, 1893) | France | | LC682685 | LC682706 | LC682728 | W. Sato-Okoshi et al. (2022, unpublished data) |

| Classification by *Blake, Maciolek & Meißner (2020)* | Classification by *Wang et al. (2022)* | Genus | Species | Locality | Museum registration number | Accession number | | | Reference |
|---|---|---|---|---|---|---|---|---|---|
| | | | | | | **18S** | **28S** | **16S** | |
| | | *Microspio* | *Microspio granulata* Blake & Kudenov, 1978 | Australia | | KP636515 | – | KP636514 | *Meißner & Götting (2015)* |
| | | *Polydora* | *Polydora cornuta* Bosc, 1802 | Japan | | LC541483 | LC541485 | LC541484 | *Abe & Sato-Okoshi (2020)* |
| | | | *Polydora hoplura* Claparède, 1868 | Japan | | LC101841 | LC101854 | LC101870 | *Sato-Okoshi et al. (2017)* |
| | | | *Polydora onagawaensis* Teramoto, Sato-Okoshi, Abe, Nishitani & Endo, 2013 | Japan | | AB691768 | LC682719 | LC595745 | *Teramoto et al. (2013)*, *Abe & Sato-Okoshi (2021)*, W. Sato-Okoshi et al. (2022, unpublished data) |
| | | *Polydorella* | *Polydorella dawydoffi* Radashevsky, 1996 | Vietnam | | – | MG460975 | MG460900 | *Radashevsky et al. (2020b)* |
| | | *Pseudopolydora* | *Pseudopolydora paucibranchiata* (Okuda, 1937) | Japan | | LC019991 | LC019995 | LC595758 | *Abe, Kondoh & Sato-Okoshi (2016)*, *Abe & Sato-Okoshi (2021)* |
| | | | *Pseudopolydora tsubaki* Simon, Sato-Okoshi & Abe, 2017 | Japan | | AB973929 | AB973937 | LC107857 | *Simon, Sato-Okoshi & Abe (2019)* |
| | | *Pygospio* | *Pygospio elegans* Claparède, 1863 | Russia | | KJ747074 | KJ747064 | KJ747084 | *Radashevsky et al. (2016b)* |
| | | | *Pygospio* sp. VVP-2014 | USA | | KJ747077 | KJ747067 | KJ747087 | *Radashevsky et al. (2016b)* |
| | | *Spio* | *Spio filicornis* (O. F. Müller, 1776) | Greenland | | FR823431 | – | FR823436 | *Meißner, Bick & Bastrop (2011)* |
| | | | *Spio* sp. 2573 | Russia | | KT200135 | KT200143 | KT200126 | *Radashevsky et al. (2016b)* |
| Sabellidae (Outgroup) | Sabellidae (Outgroup) | *Amphicorina* | *Amphicorina mobilis* (Rouse, 1990) | Japan/Australia | | AB646767 | AB646766 | HM800966 | *Yoshihara et al. (2012)*, *Capa et al. (2011)* |
| | | *Sabella* | *Sabella pavonina* Savigny, 1822 | -/Sweden/France | | U67144 | AY612632 | AY340482 | S. Nadot & A. Grant (1996, unpublished data), *Persson & Pleijel (2005)*, *Rousset et al. (2007)* |

respectively (File S1). A phylogenetic tree was constructed based on the concatenated sequences of the 18S, 28S, and 16S rRNA gene regions using maximum likelihood (ML) analyses performed with IQ-TREE (*Nguyen et al., 2015*) implemented in PhyloSuite v.1.2.2 (*Zhang et al., 2020*) under an edge-linked partition model. The TNe+I+G4, TIM3+F+I +G4, and TIM2+F+I+G4 models were selected as the best substitution models for the 18S, 28S, and 16S rRNA gene regions, respectively, by ModelFinder (*Kalyaanamoorthy et al., 2017*) as implemented in IQ-TREE under the Bayesian information criterion (BIC). We evaluated the robustness of the ML trees using the Shimodaira–Hasegawa–like approximate likelihood-ratio test (SH-aLRT) with 5,000 replicates (*Guindon et al., 2010*),

the approximate Bayes (aBayes) test (*Anisimova et al., 2011*), and ultrafast bootstraps (UFBoot) with 5,000 replicates (*Hoang et al., 2018*). An SH-aLRT ≥ 80%, aBayes ≥ 0.95, and UFBoot ≥ 95% were defined as robust statistical supports.

## RESULTS

*Systematics*

Family Spionidae Grube, 1850
Genus *Atherospio Mackie & Duff, 1986*
Type-species: *Atherospio disticha Mackie & Duff, 1986*

**Diagnosis (Emended from *Meißner & Bick, 2005*).** Prostomium deeply incised, longer than wide, posteriorly tapered and extended into a short caruncle; occipital antenna present or absent or minute process at the position of this antenna present. Nuchal organs small or indistinct. Dorsal branchiae from chaetiger 7; branchiae with distal digitiform process, outer branchial margin completely fused with notopodial postchaetal lamella. Parapodia biramous with well developed postchaetal lamellae and alimbate mostly hirsute capillaries in noto- and neuropodia. Chaetigers 4 and 5 or only chaetiger 5 with modified chaetae in the neuropodium being falcate and pointed or aristate spines, modified chaetae in regular or irregular short vertical rows superior to several capillary chaetae. Neuropodial hooks alongside capillaries; hooks uni-or bidentate, secondary tooth below main fang; hook distally with closely applied sheath. Notopodial hooks absent. Posterior spine-like notochaetae present or absent. Sabre chaetae absent but several capillaries in inferiormost position throughout the body. Genital pouches absent. Pygidium surrounded by several pairs of lateral cirri.

**Remarks.** The morphology of the new species described below is generally consistent with the diagnosis for the genus *Atherospio* by *Meißner & Bick (2005)*. Since the description of "Dorsal branchiae on chaetiger 7 and following 4–6 chaetigers" and "Postbranchial neuropodial hooks" in the diagnosis provided by *Meißner & Bick (2005)* does not apply to the new species, we emended these parts of the diagnosis as "Dorsal branchiae from chaetiger 7" and "Neuropodial hooks", respectively, to include the new species. Following the definition of caruncle by *Wong et al. (2014)*, "a dorsal extension of the prostomium, taking the form of an elevation or a distinct crest separating the nuchal organs one from another", the posterior tapered and elongated part of the prostomium present in the known *Atherospio* species is regarded as a short caruncle. Therefore, we modified the posterior part of prostomium, which is considered "not extended into a distinct caruncle" in the diagnosis by *Meißner & Bick (2005)*, to be a short caruncle. We also emended the arrangement of the modified spines from "a irregular short row" to "regular or irregular short vertical rows" since the modified spines are in very regular vertical rows in *A. disticha* and *A. aestuarii* **sp. nov.** and two forms of spines gathered in irregular, short rows in *A. guillei*. We added to the diagnosis that posterior needle-like notochaetae are found in *A. guillei* and the new species.

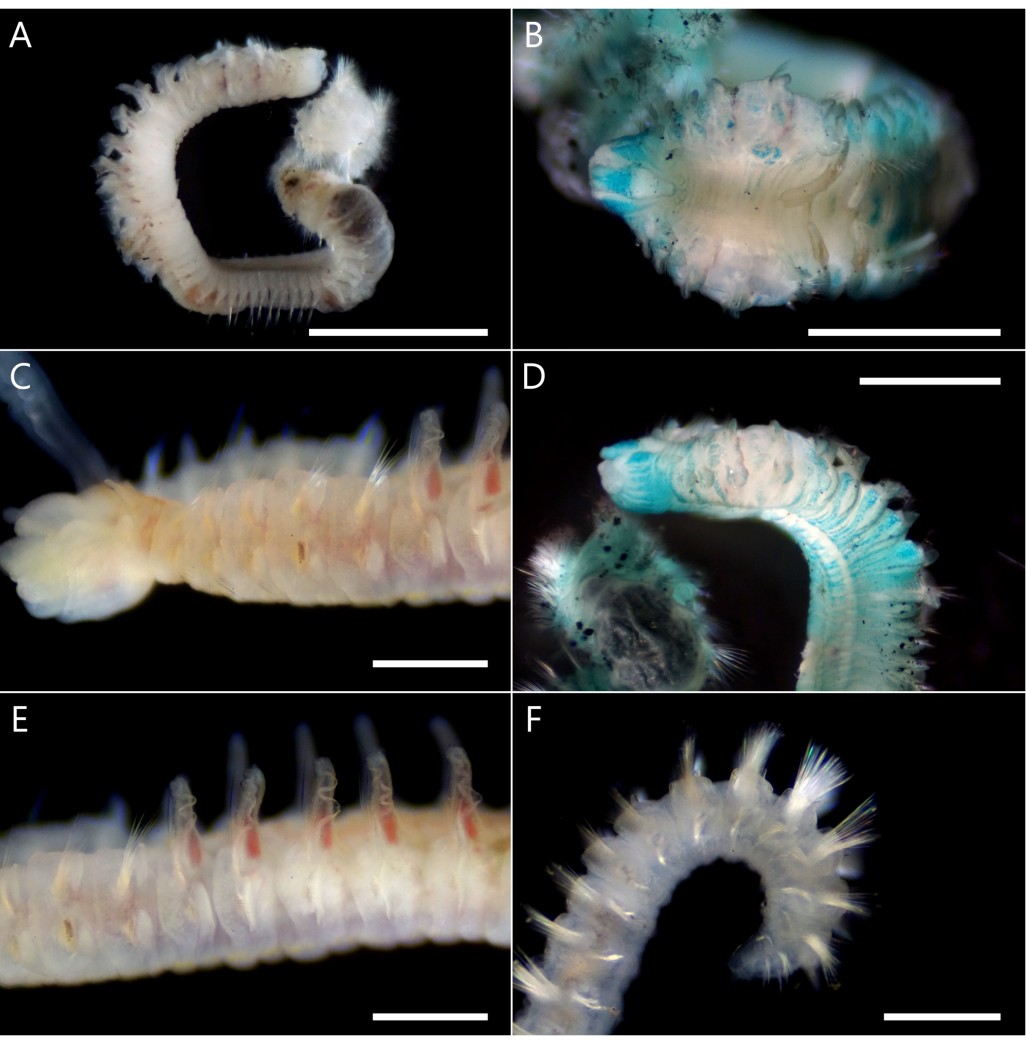

**Figure 3** *Atherospio aestuarii* **sp. nov. Light micrographs showing the morphology of preserved (A, B, D) and live (C, E, F) specimens (holotype: NSMT-Pol H-858).** (A) Entire body. (B) Anterior chaetigers, dorsal view (methyl green stained). (C) Anterior chaetigers, lateral view. (D) Anterior chaetigers, lateral view (methyl green stained). (E) Chaetigers 4–11, lateral view. (F) Posterior end, lateral view. Scale bars: (A) = 2 mm; (B, D) = 1 mm; (C, E, F) = 500 μm.

*Atherospio aestuarii* **sp. nov.**
Japanese name: Irie-nogi-supio
LSID. urn:lsid:zoobank.org:act:287692C4-C105-41BC-8718-37C6BBE10B7C
(Figs. 3 and 4)

**Type material.** Holotype: NSMT-Pol H-858, small fishing port at the mouth of the Kurio River, 30.2741 N, 130.4214 E, Yakushima Island, Kagoshima Prefecture, subtidal, <1 m depth, muddy sand, November 6, 2021 (complete specimen). Paratypes: NSMT-Pol P-859, Otomo-ura, 38.9958 N, 141.6817 E, Hirota Bay, Iwate Prefecture, intertidal, gravelly muddy sand, August 6, 2017 (incomplete 1 specimen); NSMT-Pol P-860, Otomo-ura, 38.9958 N, 141.6817 E, Hirota Bay, Iwate Prefecture, intertidal, gravelly muddy sand,

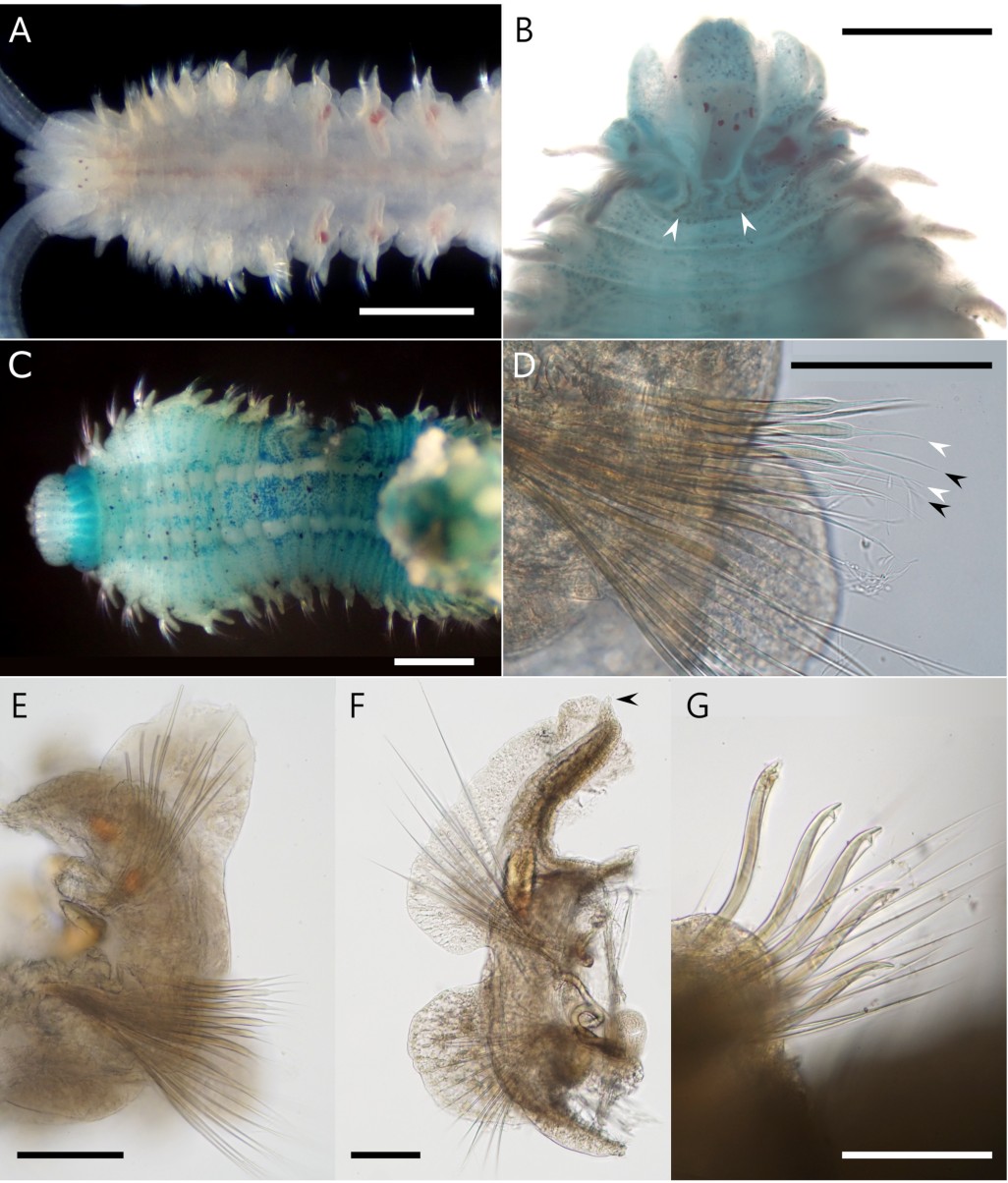

**Figure 4** *Atherospio aestuarii* **sp. nov. Light micrographs showing the morphology of living (A) and fixed (B–G) specimens (paratypes).** (A) Anterior chaetigers, dorsal view (NSMT-Pol P-866). (B) Anterior chaetigers, dorsal view (methyl green stained, NSMT-Pol P-862), arrowheads indicate the nuchal organs. (C) Anterior chaetigers, ventral view (methyl green stained, NSMT-Pol P-862). (D) Neurochaetae in left parapodium from chaetiger 5, anterior view (NSMT-Pol P-866), black and white arrowheads indicate the aristate spines in the anterior and posterior row, respectively. (E) Left parapodium from chaetiger 5, anterior view (NSMT-Pol P-866). (F) Right parapodium from chaetiger 7, anterior view (NSMT-Pol P-866), arrowhead indicates the digitiform process at the distal end of the branchia. (G) Neuropodial hooded hooks from chaetiger 34 (NSMT-Pol P-860). Scale bars: (A, C) = 500 µm; (B) = 300 µm; (D–G) = 10 µm.

August 18, 2019 (incomplete 1 specimen); NSMT-Pol P-861, Otomo-ura, Hirota Bay, 38.9958 N, 141.6817 E, Iwate Prefecture, intertidal, gravelly muddy sand, August 4, 2020 (incomplete 1 specimen); NSMT-Pol P-862 (incomplete 1 specimen), NSMT-Pol P-863

(incomplete 7 specimens), NSMT-Pol P-864 (incomplete 1 specimen), nameless small inlet in Ago Bay, 34.2985 N, 136.8311 E, Mie Prefecture, subtidal, <1 m depth, gravelly muddy sand, October 8, 2021; NSMT-Pol P-865 (incomplete 1 specimen), NSMT-Pol P-866 (incomplete 2 specimens), small fishing port at the mouth of the Kurio River, 30.2741 N, 130.4214 E, Yakushima Island, Kagoshima Prefecture, subtidal, <1 m depth, muddy sand, November 6, 2021.

**Description.** Holotype complete (pygidium damaged) with 64 chaetigers, measuring 9.5 mm long and 1.2 mm wide at chaetiger 5 (Fig. 3); paratypes incomplete up to 14.4 mm long, 1.5 mm wide for 40 chaetigers. Body wide, dorsoventrally flattened for first six chaetigers (Figs. 3B and 4A), then gradually narrower and becoming cylindrical in cross-section. Body white to light tan in preserved specimen (Fig. 3A), translucent white to light tan when alive with red blood vessels and pale orange to brown digestive tract internally (Figs. 3C, 3E and 4A); body and palp pigmentation absent.

Prostomium longer than wide, anteriorly incised; extends posteriorly as caruncle to middle of chaetiger 1 (Figs. 3B and 4A). Eyes dark red, two pairs arranged in trapezoidal shape, lateral pair situated anteriorly, kidney-shaped, larger than medial ones (Figs. 4A and 4B). Occipital antenna absent. Nuchal organs U-shaped with outward curving posterior part, located just behind prostomium and between notopodial lamellae of chaetiger 1 (Figs. 3B and 4B). Palps arising lateral to prostomium (Fig. 4A). Peristomium extending lateral to prostomium, forming upper lip of mouth and extending ventrally forming ventral lip of mouth; thick everted proboscis or pharynx present; oral lips relatively smooth; peristomial papillae (see *Blake & Maciolek, 2018*) absent.

Chaetigers 1–6 abranchiate (Figs. 3B, 3C and 4A). Notopodial postchaetal lamellae long, digitiform or lanceolate on chaetiger 1 (Figs. 3B and 4C), broader on chaetiger 2, and becoming broad triangular or oval on chaetigers 3–6 (Fig. 4E). Neuropodial lamellae digitiform or lanceolate on chaetiger 1, broad triangular on chaetiger 2, and oval to triangular on chaetigers 3–6. Chaetiger 5 of same size as neighboring chaetigers. Midventral series of white rectangular pads in anterior chaetigers, indistinct in fixed specimens.

Branchiae from chaetiger 7 to 18–23 (File S2), long and cirriform, with digitiform process at distal end (Fig. 4F); overlapping mid-dorsal or not, full-sized from chaetigers 10–12; fully fused with notopodial postchaetal lamellae in outer margin (Fig. 4F); ciliation along inner margin, extending to a transverse ciliated band across the whole width of the chaetiger. In branchial chaetigers, notopodial postchaetal lamellae foliated and often wavy, especially when alive (Figs. 3C and 3E); neuropodial postchaetal lamellae rounded, larger dorsoventrally than that of chaetigers 1–6 (Fig. 4F). In postbranchial chaetigers, both postchaetal lamellae smaller, rather more subtriangular.

Notochaetae in most chaetigers long slender capillaries without limbations; some posterior notopodia with bundles of needle-like capillaries raised dorsally (Fig. 3F); notopodial hooks absent. Neurochaetal capillaries without limbations in anterior chaetigers. Neuropodia of chaetiger 5 double vertical rows of aristate spines dorsal to small bundle of capillaries; spines in posterior row slightly thicker than those of closely applied

**Table 2 Taxonomic characteristics of three species in *Atherospio Mackie & Duff, 1986*.**

| Character | Species | | |
|---|---|---|---|
| | ***A. disticha Mackie & Duff, 1986*** | ***A. guillei (Laubier & Ramos, 1974)*** | ***A. aestuarii* Abe & Kan, sp. nov.** |
| Prostomium: anterior margin | 2 rounded lobes | 2 lobes, deeply incised | 2 lobes, deeply incised |
| Occipital antenna | Short | Absent | Absent |
| Peristomial papillae | Not reported | Present | Absent |
| Anterior notopodial lamellae[1] | 1–2: digitiform; 3–6: broad, triangular | 1: digitiform; 3–6: broad, triangular | 1: digitiform; 3–6: broad, triangular or oval |
| Anterior neuropodial lamellae[1] | 1–2: broad, triangular; 3–6: elliptical | 1: digitiform; 3–6: broadly rounded | 1: digitiform; 3–6: oval to triangular |
| Branchial distribution[1] | 7 to 11/12: broad, fully fused to dorsal lamellae | 7 to 11–13: long, thick, fully fused to dorsal lamellae | 7 to 18–23: long, thick, fully fused to dorsal lamellae |
| Modified anterior neurochaetae | Chaetigers 4–5 with double vertical row of aristate spines | Chaetiger 5 with 2–3 heavy spines and 3+ thin spines | Chaetiger 5 with double vertical row of aristate spines |
| Posterior neuropodial hooks | Bidentate hooded hooks with narrow, curved shaft from chaetiger 13–15 | Uni- and bidentate with straight or curved shaft; hood absent; from chaetiger 15–16 | Bidentate hooded hooks with narrow, curved shaft from chaetiger 16–19 |
| Posterior needle-like notochaetae | Absent | Present | Present |
| Pygidium | 6–9 cirri | 8 cirri | Unknown |
| Methyl green staining | Not tested | No pattern | Prostomium, peristomium, and posterior to 7th chaetiger are clearly stained |
| Distribution | West coast of Scotland: 27 m, Celtic Deep: >100 m, Kattegat: 50 m | North Sea: 38–41 m, Mediterranean Sea: 44–99 m | Japan, intertidal to subtidal shallower than 1 m depth |
| References | *Mackie & Duff (1986)*, *Mackie, Oliver & Rees (1995)* | *Laubier & Ramos (1974)*, *Meißner & Bick (2005)* | This study |

**Note:**
[1] Numbers refer to the chaetigers on which the character appears.

anterior row, taper steeply towards tip with short aristae part; spines in the anterior row taper gradually towards tip with long aristae part (Figs. 4D and 4E). Hooded hooks in neuropodia from chaetigers 16–19 (File S2) to the posterior-most chaetiger, accompanied by capillaries in all chaetigers; numbering up to 6 in a series, reduced in posterior chaetigers, shafts S-curved, hooks bidentate with secondary tooth on concave side at right angle to and below main fang (Fig. 4G). Neuropodial sabre chaetae absent.

Pygidium without anal cirri probably due to damage.

**Methyl green staining.** Anterior half of the prostomium deeply stained (Fig. 3B). Peristomium stained with vertical stripes (Figs. 3D and 4C). Tips of some post-chaetal lamellae deeply stained. Chaetigers 1–6 diffusely stained with scattered deeply stained cells on both dorsal and ventral sides; chaetiger 7 onward more strongly stained than chaetigers 1–6 on dorsal ventral, and lateral sides (Figs. 3B, 3D and 4C). Unstained ventral large white spots, one pair per chaetiger, present from chaetiger 2 to posterior middle-body chaetigers (Fig. 4C).

**Remarks.** *Atherospio aestuarii* **sp. nov.** closely resembles *A. disticha* and *A. guillei* and is intermediate in morphology between these species. *Atherospio aestuarii* **sp. nov.** is similar

to *A. disticha* and differs from *A. guillei* in having branchiae fused to the notopodial lamellae on a restricted number of segments from chaetiger 7, modified neurochaetae on chaetiger 5, and at least some bidentate neuropodial hooks with the secondary tooth below the main fang (Table 2). The form and arrangement of the modified aristate neurochaetae in double vertical rows closely resemble those found on chaetigers 4 and 5 of *A. disticha*. The new species lacks the occipital antenna present in *A. disticha*. In this respect it resembles *A. guillei*, however, that species differs in having robust neuropodial spines on chaetiger 5 and peristomial papillae (see *Meißner & Bick, 2005*: fig. 2C), and a preponderance of unidentate neurochaetae. Both *A. guillei* and the new species have slender needle-like notochaetae in their posteriormost chaetigers. *Atherospio aestuarii* **sp. nov.** is distinguished from both congeneric species by its branchial and neuropodial hook distributions; as the last branchial chaetiger and the first chaetiger with neuropodial hook are more posterior in the former species. The other two nominal *Atherospio* species were collected from ≥27 m depths in the subtidal zone (Table 2), whereas the new species was unique in that it was recorded at relatively shallow depths, which included intertidal zones.

*Mackie, Oliver & Rees (1995)* and *Mackie & Garwood (1995)* reported two provisionally unnamed spionid taxa closely related to *A. disticha* from Cardigan Bay in the Irish Sea as 'Spionidae gen. A' and 'Spionidae gen B' and mentioned that 'Spionidae gen. B' is morphologically similar to *A. guillei* (as *Polydora*). Several *Atherospio* related taxa collected from Europe and Hong Kong including 'Spionidae gen. A' and 'Spionidae gen B' were referred as 'Genus A' and 'Genus B' (include *A. guillei*, but may also involve two separate taxa) in *Mackie (1996)*. In his character matrices which provided the main characteristics of the morphology of these two groups (*Mackie, 1996*: Tables 2 and 3), 'Genus A' and 'Genus B' are distinguished by the former lacking and the latter having posterior modified notochaetae and the former having one type of anterior modified neurochaetae but the latter having two types. *Atherospio aestuarii* **sp. nov.** does not fall into either group because it has posterior needle-like notochaetae and one type of anterior modified neurochaetae.

**Etymology.** The specific name *aestuarii* is from the Latin word *aestuarium*, which means the estuary, inlet, and intertidal zone, thus referring to the habitat of this species.

**Habitat.** Muddy and gravelly muddy sand sediment in the intertidal to subtidal zone, <1 m in depth.

**Distribution.** Currently identified in Otomo-ura, Hirota Bay (Iwate Prefecture), Ago Bay (Mie Prefecture), and Yakushima Island (Kagoshima Prefecture), Japan.

*Molecular phylogeny*

The intraspecific p-distances in the 18S, 28S, and 16S rRNA gene sequences of the seven *A. aestuarii* **sp. nov.** specimens were 0%, 0–0.26%, and 0–2.20%, respectively. In the molecular phylogenetic analyses based on the concatenated sequences, the *Polydora* complex + *Pygospio*, subfamily Spioninae *sensu Blake, Maciolek & Meißner (2020)* + *Glandulospio*, and that plus *Marenzelleria* Mesnil, 1896, *Rhynchospio* Hartman, 1936, *Atherospio, Dispio* Hartman, 1951, *Scolelepis*, and *Malacoceros* were recovered as clades

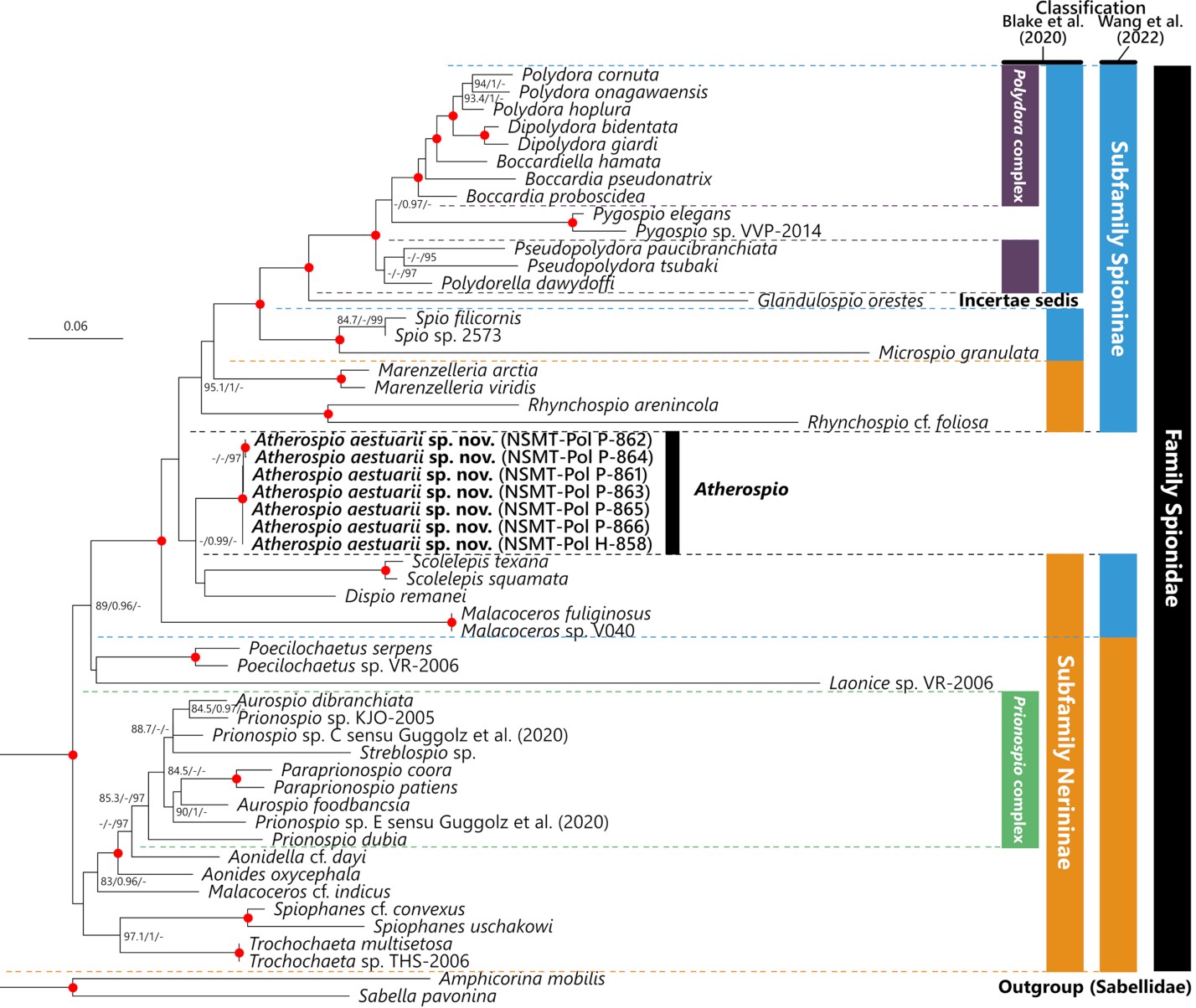

**Figure 5** **Maximum likelihood tree inferred from concatenated sequences of nuclear 18S and 28S and mitochondrial 16S rRNA gene sequences of spionid species obtained in the present study and from the DDBJ/EMBL/GenBank database (Table 1).** The gene sequences obtained in this study are highlighted in boldface. The organism names of unidentified species are labeled with the identifiers in the DDBJ/EMBL/GenBank database. The subfamily classifications defined by *Blake, Maciolek & Meißner (2020)* and *Wang et al. (2022)* are shown in the colored bars on the right side and black, blue, red, green, and yellow bars indicate the family Spionidae, subfamilies Spioninae and Nerininae, *Polydora* complex, and *Prionospio* complex, respectively. SH-aLRT/approximate Bayes support/ultrafast bootstrap support values of ≥80%/≥0.95/≥95%, respectively are given beside the respective nodes. Nodes with red circles indicate triple high support values of SH-aLRT ≥ 80, approximate Bayes support ≥0.95, and ultrafast bootstrap support ≥95. The scale bar represents the number of substitutions per site. Sequences of *Amphicorina mobilis* and *Sabella pavonina* are used for outgroup rooting.                                           

with robust statistical support (SH-aLRT ≥ 80%, aBayes ≥ 0.95, UFBoot ≥ 95%) (Fig. 5). *Atherospio aestuarii* **sp. nov.** formed a clade with *Dispio* and *Scolelepis*, however, the support value for the node was not robust (SH-aLRT = 76.3, aBayes = 0.99, UFBoot = 54).

## DISCUSSION

In contrast to the results of phylogenetic analyses by *Sigvaldadóttir, Mackie & Pleijel (1997)* and *Blake & Arnofsky (1999)*, our molecular phylogenetic analysis indicated that *Atherospio aestuarii* **sp. nov.** did not form a clade distinct from the subfamilies Spioninae and Nerininae, but rather could be included within a clade that included the genera of the subfamily Spioninae *sensu Blake, Maciolek & Meißner (2020)* plus *Glandulospio*, *Marenzelleria*, *Rhynchospio*, *Scolelepis*, *Dispio*, and *Malacoceros* (Fig. 5). This clade corresponds to that referred to as the subfamily Spioninae in the alternative classification of the subfamily suggested by *Wang et al. (2022)* based on the results of molecular phylogenetic analysis. Monophyly of Spioninae *sensu Wang et al. (2022)* was supported also by *Abe & Sato-Okoshi (2021)* and the present study. However, the alternative subfamily classification suggested by *Wang et al. (2022)* has the following problems: (1) Nerininae *sensu Wang et al. (2022)* has been recovered as either monophyletic with low support (*Wang et al., 2022*) or as paraphyletic (*Abe & Sato-Okoshi, 2021*, this study) and (2) if Nerininae does not include *Scolelepis*, then this subfamily is not valid because the type-genus is *Nerine* which is a junior synonym of *Scolelepis*. The paraphyly of Nerininae *sensu Blake, Maciolek & Meißner (2020)* is also clearly indicated by the previous (*Abe & Sato-Okoshi, 2021*; *Wang et al., 2022*) and the present study. Our understanding of the phylogenetic relationships among the spionid genera is still in a precocious phase, and therefore the subfamily classification of the Spionidae should be revisited with more comprehensive and robust molecular phylogenetic tree as well as non-molecular data such as morphology, development, and reproduction about representatives of many genera. Nevertheless, our molecular phylogenetic analysis supports previous recognitions by *Mackie & Duff (1986)*, *Radashevsky & Fauchald (2000)*, and *Radashevsky (2012)* which indicate that the members belonging to the *Pygospiopsis-Atherospio* group are not closely related to the superficially similar taxa, that is, *Polydora* and *Pygospio*, and that the heavy spines in the fifth segments of *Polydora* and *Atherospio* are not homologous *sensu* stricto. The possibility of a close relationship between *Atherospio* and *Dispio*/*Scolelepis* is worth further investigation through molecular phylogenetic analysis with the addition of potential closely related taxa such as *Australospio* Blake & Kudenov, 1978 (*Sigvaldadóttir, Mackie & Pleijel, 1997*), *Lindaspio* Blake & Maciolek, 1992 (*Mackie, 1996*), and *Pygospiopsis*, as the statistical support for the clade was not robust in the present study.

## ACKNOWLEDGEMENTS

We would like to express our sincere appreciation to Taeko Kimura (Mie University) for giving us the opportunity to conduct preliminary observations of the specimens from Ago Bay; Masatoshi Matsumasa (Iwate Medical University), Takao Suzuki (Michinoku Research Institute for Benthos), Kyoko Kinoshita (Tohoku University), Takashi Inoue, and Hiroka Hidaka (Japan Wildlife Research Center) for their help with fieldwork in Otomo-ura; Genki Kobayashi (Kyoto University) for his help with fieldwork in Ago Bay and Yakushima Island; Naoto Jimi (Nagoya University) and the staff of Sugashima Marine Biological Station for their assistance and for allowing us to use their laboratory during our

survey in Ago Bay; Andrew S.Y. Mackie, Vasily I. Radashevsky, and an anonymous reviewer for their helpful comments and suggestions in the peer review process. Sample collection in Otomo-ura in 2017 was conducted as a project of "The Ecosystems Monitoring Survey of the Pacific Coastal Areas of the Tohoku Region" by the Biodiversity Center of Japan, Nature Conservation Bureau, Ministry of the Environment.

### Funding

This work was supported by JSPS KAKENHI (Grant Number: JP19K15899), the Environment Research and Technology Development Fund (Grant Number: JPMEERF20204R01) of the Environmental Restoration and Conservation Agency of Japan, and the Biodiversity Conservation Research Activity Promotion Project of the Yakushima Environmental and Cultural Foundation. The funders had no role in study design, data collection and analysis, decision to publish, or preparation of the manuscript.

### Grant Disclosures

The following grant information was disclosed by the authors:
JSPS KAKENHI Grant Number: JP19K15899.
Environment Research and Technology Development Fund of the Environmental Restoration and Conservation Agency of Japan Grant Number: JPMEERF20204R01.
Biodiversity Conservation Research Activity Promotion Project of the Yakushima Environmental and Cultural Foundation.

### Competing Interests

The authors declare that they have no competing interests.

### Author Contributions

- Hirokazu Abe conceived and designed the experiments, performed the experiments, analyzed the data, prepared figures and/or tables, authored or reviewed drafts of the article, sample collection, and approved the final draft.
- Kotaro Kan performed the experiments, authored or reviewed drafts of the article, sample collection, and approved the final draft.

### Field Study Permissions

The following information was supplied relating to field study approvals (*i.e.*, approving body and any reference numbers):

The water areas where the specimens were collected in this study are not protected, and no permission of any kind is required to collect the organisms. In the field survey of this study, we did not collect any commercially marine species and did not use any collection method that violated the prefectural fishery regulation, so we did not need any permission for the survey.

## DNA Deposition

The following information was supplied regarding the deposition of DNA sequences:

All the sequences newly generated in this study are available at the DDBJ/ENA/GenBank nucleotide sequence databases: LC685029 to LC685049.

## Data Availability

The raw data are available in the Supplemental Files.

## New Species Registration

The following information was supplied regarding the registration of a newly described species:

Publication LSID: urn:lsid:zoobank.org:pub:ED1D54BF-7C4E-4277-A675-F604C743E6C7

*Atherospio aestuarii* sp. nov. LSID: urn:lsid:zoobank.org:act:287692C4-C105-41BC-8718-37C6BBE10B7C

## Supplemental Information

Supplemental information for this article can be found online at http://dx.doi.org/10.7717/peerj.13909#supplemental-information.

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
