# Peer review of "Phylogenetic position of the enigmatic genus Atherospio and description of Atherospio aestuarii sp. nov. (Annelida: Spionidae) from Japan"

_PeerJ, doi:10.7717/peerj.13909_

## Round 0.1 · original submission · Major Revisions

Dear Author, please find attached three comprehensive reviews of your ms; although the reviewers were complimentary of your work they have also suggested many improvements. Please pay particular attention to their comments on the phylogenetic analysis, and its taxonomic implications, and make the appropriate adjustments. If you disagree with any particular points, please provide a counterargument. I think the figures, in general, are fine as is. Looking forward to your revised version. Best Chris

Reviewer 1 ·

Basic reporting

I think this is a useful paper in the part describing a new species of the truly enigmatic genus Atherospio, whose phylogenetic relationships remain uncertain. The descriptive part of the manuscript is well written and does not need major revision, although in many places English editing and spelling should be done.

Experimental design

My major concern is about phylogenetic analysis and the conclusions drawn from it. The authors have stated (see Lines 178-180) that “The final lengths of the aligned sequences were 1703, 663, and 434 bp for the 18S, 28S, and 16S rRNA gene sequences, respectively.” However, the only available 18S sequence of Glandulospio orestes, which is included in the analysis, has 402 bp, and the only available sequences of Dispio have 671 bp for 18S (KU900474) and 346 bp for 28S (KU900467). The situation is similar for many other principal taxa included in the analysis (see Supplementary_file_S1) and whose phylogenetic relationships are discussed by the authors. In my opinion, only sequences of the same length can be included in the analysis. I suggest the authors to include in the analysis only sequences of the same length.

Validity of the findings

As a result of including inappropriate data in molecular analysis, the whole Discussion section appears rather speculative.

Additional comments

Lines 37-39. This is a confusing statement. I suggest to reword it:
“The results of our molecular phylogenetic analysis indicate that the new species did not form a clade distinct from the subfamilies Spioninae or Nerininae but was included within either subfamily”

Lines 42-44. This is a confusing statement. Sister relationships cannot be between three taxa. “We also point out for the first time the possibility of a close relationship between Atherospio and Dispio Hartman, 1951 and Scolelepis Blainville, 1828 as they showed sister relationships in the phylogenetic tree”

Line 67. “Blake et al. (2019)” This book was printed in 2020 and should be cited correspondingly.

Line 69. Soderstrom should be written as Söderström.

Lines 142-151. I don’t think that these statements are really needed. At least not in this place, not in the section “Morphological observation” The Line 222 provides all required information.

Lines 178-180. “The final lengths of the aligned sequences were 1703, 663, and 434 bp for the 18S, 28S, and 16S rRNA gene sequences, respectively.” Here is my major concern about analysis of molecular data – using very short sequences earlier obtained by other authors. For example, the only available 18S sequence of Glandulospio orestes has 402 bp, and the only available sequences of Dispio have 671 bp for 18S and346 bp for 28S. The situation is similar for many other taxa included in the analysis.

Line 207. Why only postbranchial? “Postbranchial neuropodial hooks alongside capillaries”

Lines 239-.. The telegraph style should be applied consistently throughout Description section.

Lines 245-246. I see a conflict between the diagnosis of Atherospio on lines 199-200: “Prostomium… posteriorly tapered and not extended into a distinct caruncle”, the description of A. aestuarii on lines 245-246. “Prostomium … extending to middle of chaetiger 1” and line 248 saying: “Caruncle… absent”. My question is How do you define the caruncle?

Line 252. Can you explain what is this? “peristomial papillae”

Line 259. I am not convinced that large white indistinct spots of figure 4C represent “Ventral epidermal glands” as interpreted by the authors. Better photo of higher magnification id needed to confirm the authors’ opinion.

Line 264. “transverse ciliated band across the whole width of the chaetiger” These bands are called nototrochs in spionids and other polychaetes.

Lines 312-313. Please explain the values in brackets: “0% (0/1780 bp), 0%–0.26% (0–2/769–775 bp), and 0%–313 2.20% (0–11/501–503 bp)”
1) I cannot understand what are the values in front of slash /
2) Why the ranges are shown for the bp numbers? The p-distances should be calculated for sequences of the same length.

Lines 323-324. “Atherospio aestuarii sp. nov. did not form a clade” Yes, one species cannot form a clade. I suggest that you improved this part of the text.

Lines 323-327. “the results of our molecular phylogenetic analysis indicated that Atherospio aestuarii sp. nov. did not form a clade distinct from the subfamilies Spioninae or Nerininae, but rather could be included within either subfamily (Fig. 5). This implies other members of the Pygospiopsis-Atherospio group may show similar phylogenic positions as well as the new species.” This is a very confusing speculation. I suggest to avoid speculations of this kind and discuss only the topology obtained in the present analysis.

Line 327. “Following the classification of the subfamily by Blake et al. (2019” Blake et al. and others based their classifications on different sets of characters and arguments. Therefore, you cannot follow their classifications.

Lines 330-331, 338-339. “Recently, Wang et al. (2022) suggested an alternative classification” … “Following this classification, our results classified Atherospio within Spioninae” I think that this is incorrect reasoning, which cannot be used for systematic inferences.

Line 621. “Stereo- (A, C) and light micrographs” This is a very unusual opposition of “stereo” and “light” micrographs. Are not all of them taken with light microscopes?

Some minor comments are given directly in the text (PDF attached).

Annotated reviews are not available for download in order to protect the identity of reviewers who chose to remain anonymous.

Reviewer 2 ·

Basic reporting

This paper is intended to describe a new species of the rare and poorly known spionid genus Atherospio from Japan, the first species of this genus to be reported outside of the Atlantic Basin. The species belongs to an unusual group of spionids having unusual branchial patterns and a unique kind of hooded hook. The authors do provide a description of this species and carefully compare it with the two previously known species and justify it as a different and new species. A table is presented that compares the 3 species of Atherospio.

However, the figures do not adequately illustrate the main characters. Photographs are used that are not of sufficient resolution to see details of the pre-chaetiger region, branchial shape distribution, and details of the spines on chaetiger 5. I suggest that SEMs or good drawings would be preferred if better photographs cannot be taken.

However, although the new species is described and differentiated adequately from its relatives, the authors have spent much of the paper discussing (hand-waving) a poorly conceived "phylogenetic" analysis which they admit is not well supported statistically. In other words, the results they have submitted are not statistically significant. Yet pages of text are devoted to a flawed result.

There are comments included in the text itself that deal with this flawed approach. If this "phylogeny" is supported, then the authors have removed the genus Scolelepis from the subfamily Nerininae to the subfamily Spioninae. They do not seem to understand that since the Nerininae Soderstrom, 1920 is based the type genus Nerine which is a junior synonym of Scolelepis, then the subfamily Nerininae becomes invalid if it does not include species of Scolelepis. Although subfamilies are not currently recognized in the World Register of Marine Species, it still behoves the authors to understand some basic taxonomy of the names being used.

Experimental design

The only experimental design is the nature of the "phylogenetic" analysis. The authors appear to be based their analysis on several recent studies by the same authors. As noted above the results presented were noted as not statistically significant. Rather than attempting other approaches, such as using only the nuclear genes and comparing the results, they devoted considerable text to discussing a flawed result. Also, the COI gene has not been sequenced, which is now the so-called bar code gene which can also be used in comparing related genera and species.

Validity of the findings

The discovery of and description of the new species is well done, although the illustrations need improvement. However, as noted the emphasis on a flawed analysis of the molecular data and an effort make it into something important detracts from the basic discovery of a new species.

Additional comments

No further comment.

Annotated reviews are not available for download in order to protect the identity of reviewers who chose to remain anonymous.

·

Basic reporting

This is a very interesting MSS to me and is generally well written; a good read. There are a few parts that need some rewording for clarity and accuracy, and some typographic errors. However, the overall MSS is good and the new molecular analysis welcome. It is an important contribution to understanding the Pygospiopsis-Atherospio group.

There is one omitted reference that should be consulted:

Mackie ASY. 1996. Taxonomy and phylogeny of spioniform polychaetes (Annelida). Ph.D. Thesis, Göteborgs Universitet, 175 pp. Available at https://www.researchgate.net/publication/230608224_Taxonomy_and_phylogeny_of_spioniform_polychaetes_Annelida

This publication includes a phylogenetic analysis of Spionidae, including ‘Genus A’ (= animals of similar morphology to Atherospio aestuarii, and ‘Genus B’ (= animals similar to Atherospio guillei). See Chapter 1, pp. 13-14, Tables 1 & 2, and associated cladistic analyses. The publication was also cited by Meißner & Bick (2005) and Wang et al. (2022).

Although I have a preference in general for line drawings, the photographic illustrations are generally very good. The modified aristate spines of chaetiger 5 and the neuropodial hooks are shown well. I would have liked a close up of the posterior needle-like notochaetae also since their form is not entirely clear in this paper, or in that of Meißner & Bick (2005).

Experimental design

These are all appropriate and well-described. My only comment here concerns ‘Specimen Collection’ (Lines 119-129). The localities are detailed and reference is made to the maps (Fig.1) and the locality photographs (Fig. 2). Information on the habitats is given in the ‘Type material’ section (Lines 225-237). I notice that most habitats are said to be “muddy flat”. However, Figs 2A & 2B appear to show more mixed sediments than just mud. Perhaps more precise information could be given? e.g., muddy sand with shell, gravelly muddy sand, etc. as appropriate.

Validity of the findings

The new species is well justified, though the comparisons with Atherospio disticha and A. guillei are a little muddled and could be made much clearer (Lines 30-35 & Lines 289-297).

For example (Lines 30-35): “in having branchiae fused to the notopodial lamellae on a restricted number of segments from chaetiger 7, modified neurochaetae on chaetiger 5, and at least some bidentate neuropodial hooks with the secondary tooth below the main fang. The form and arrangement of the modified aristate neurochaetae in double vertical rows closely resemble those found on chaetigers 4 and 5 of Atherospio disticha. The new species lacks the occipital antenna present in A. disticha. In this respect it resembles A. guillei, however, that species differs in having robust neuropodial spines on chaetiger 5, and a preponderance of unidentate neurochaetae. Both A. guillei and the new species have slender needle-like notochaetae in their posteriormost chaetigers. Atherospio aestuarii is distinguished from both congeneric species by its branchial and neuropodial hook distributions.”

The same or similar text could be used in lines 289-297. Concerning the presence of slender posterior needle-like notochaetae, it is interesting that Mackie (1996: Tables 2 & 3) recorded modified posterior notochaetae in his Genus B (= Atherospio guillei), but not in his Genus A (= animals otherwise similar to Atherospio aestuarii).

See also Mackie & Garwood (1995: 41) & Mackie et al. (1995) regarding occurrence of Genus A and B animals in the Irish Sea.

Mackie ASY, Garwood PR. 1995. 5.1 Annelida In: Mackie ASY, Oliver PG, Rees, EIS. Benthic biodiversity in the southern Irish Sea. Studies in Marine Biodiversity and Systematics from the National Museum of Wales. BIOMÔR Report 1: 37–50. Available from https://www.researchgate.net/publication/230597242_Benthic_biodiversity_in_the_southern_Irish_Sea

The Molecular phylogeny is very welcome and the results presented well in Fig. 5. The authors must be careful when discussing their resulting Maximum Likelihood Tree (Line 325-328; see also Lines 38-39) with relation to other subfamily classifications. Regarding the position of Atherospio aestuarii, the statement that it “could be included within either subfamily” (Line 325) is incorrect. The authors have included the subfamily groupings of Blake et al. (2019) and Wang et al. (2022) in Fig. 5 to show how the results of those authors correspond with theirs. However, their own results only show agreement with the second authors and hence Atherospio aestuarii can only be included in the Spioninae sensu Wang et al. The Nerininae sensu Blake et al. is not supported in Fig. 5, therefore it is NOT “plausible to place Atherospio in Nerininae” (Lines 327-328). The text from Line 325-328 should simply be omitted. The findings of Blake et al. and Wang et al., and their relationships with the new tree are better described (with some minor adjustments) in the text following these lines.

Similarly Lines 38-39. Delete “but was included within either subfamily,”

Additional comments

OTHER COMMENTS
Lines Change
68-70 Suggest insert numbers (1-4) to make the four categories clearer
e.g. … Blake (2006): 1. Subfamily Subfamily Nerininae Söderström, 1920; 2. Subfamily Spioninae Söderström, 1920; 3. Clade consisted of Pygospiopsis, Atherospio, and Pseudatherospio (= Pygospiopsis-Atherospio group); and 4. five monotypic genera

Note Söderström

80 Atherospio
86-93 See also Mackie (1996) details above
93 was subsequently deemed unfortunate
133 Question: was 70% (not 100%) ethanol used for the molecular analyses?
153-191 Comment: I have no expertise with molecular work, so cannot comment on this section.
207 Comment: Postbranchial is incorrect since in the new species the neuropodial hooks almost always start before the end of the branchial region (see text and Supplementary file S2)
208 ADD: posterior spine-like notochaetae present or absent
212-217 Explain the above two changes to the Diagnosis
225-237 See above regarding sediment descriptions
239 Holotype complete (pygidium damaged)
254-255 Notopodial postchaetal lamellae long, digitiform or lanceolate

Note: digitiform is a more accurate term to use than digitate here. Change elsewhere also

267-268 rather more subtriangular.
270 Question: Are they “needle -like capillaries” or “needle-like spines” ?
289-296 See above regarding species distinguishing text
304 Question: were any of the specimens reproductive?
305 See previous comments about nature of the muddy sediment
325-328 See comments above regarding Fig. 5 and other classifications
340 which was supported also by Abe & Sato-Okoshi (2021) and the present study.
341 Nerininae sensu Blake et al. (2019)
362-363 Comment: but see Mackie (1996)
370 Atherospio
390: References
Check format with Instructions to authors regarding dates not being in brackets, journal titles italicised etc

458 disticha

625 (D) Neurochaetae

636 The subfamily classifications

Table 2 digitiform instead of digitate

Table 2 Comment: Consider adding another character
“Posterior needle-like notochaetae”

---

## Round 0.2 · Minor Revisions

Dear Author,

After receiving your revised manuscript, we were eventually able to secure further comments from 2 of the 3 initial reviewers (sorry for the delay); both have acknowledged improvement in the revised version, but suggest that further changes are warranted.

Also, I have provided editorial commentary in the attached pdf. Please note that the authors are not suggesting any re-analysis, but instead provide useful comments on terminology and character interpretation. I hope you find these useful and look forward to seeing your revised version.

All the best Chris

Reviewer 1 ·

Basic reporting

No comments

Experimental design

No comments

Validity of the findings

No comments

Additional comments

From the new version of the text and authors’ response to the reviewers I see that the manuscript has been carefully revised and thus fits the requirements of the journal and criticisms of the reviewers. The description and molecular data of the new species add valuable information for further investigation on the diversity and phylogeny of one of the largest and diverse polychaete family Spionidae.
My major concern is about attempt to hypothesize on the phylogenetic relationships between spionid genera using available molecular data. As I said in my previous review, the analysis performed in the present study, same as in earlier study (Abe & Sato-Okoshi 2021) and by other authors (Wang et al. 2022) are not eligible from the fundamental point of view. Having too many short sequences in the analysis (too many question marks in the concatenated alignment) creates a source for false hypotheses which can have a good node support but still be false because based on false data.
I am not satisfied with the authors’ response “Molecular phylogenetic analysis has the advantage of including sequences of different lengths in the analysis. In the present analysis, we reanalyzed the data by excluding data with short sequences, but since the node support values did not improve, we have left those sequences in the revised manuscript without removing them.” Nevertheless, I do not insist to redo or remove the phylogeny presented by the authors. This will serve as an example of our poor current knowledge of spionids and little available molecular data about these polychaetes at present. We still need both morphological and molecular data about representatives of many genera to provide a more or less stable hypothesis about their relationships. Without such data, all the current hypotheses seem precocious and therefore contradicting each other. This refers not only to the issues inferred in the analyses of molecular data mentioned above, but also to the hypotheses based on incomplete and often irrelevant morphological data (e.g. Sigvaldadottir et al. 1997; Blake & Arnofsky 1999; Blake et al. 2020).

Some little comments/suggestions are given by me below and can easily be fixed by the authors:

Lines 126-130 and on page 11. Comment: I strongly recommend to use the decimal format of the coordinates for sampling sites instead of outdated minutes, degrees and seconds.

Lines 245-246. I see a conflict between the diagnosis of Atherospio on lines 199-200: “Prostomium… posteriorly tapered and not extended into a distinct caruncle”, the description of A. aestuarii on lines 245-246. “Prostomium … extending to middle of chaetiger 1” and line 248 saying: “Caruncle… absent”. My question is How do you define the caruncle?
Reply: Caruncle is defined as a dorsal extension of the prostomium, taking the form of an elevation or a distinct crest separating the nuchal organs one from another by Wong et al. (2014). Atherospio species posteriorly tapering prostomium but do not have caruncle.
Comment: So, your description of the posterior part of the prostomium perfectly matches the definition of the caruncle: “Prostomium… posteriorly tapered and … extending to middle of chaetiger 1” Therefore, I again recommend to follow this definition and name the posterior elongated part of the prostomium as a caruncle.

Table 1 legend: … The gene sequences obtained in this study are highlighted in boldface type, together with the museum registration number of the specimens.
Comment: I do not see any sequence or museum registration number highlighted in boldface type!

Figure 5 legend: Comment: The legends should provide complete explanation for the figures. Explain what mean VVP-2014, V040, VR 2006, etc. shown on the tree.

Vasily Radashevsky

---

## Round 0.3 · accepted · Accept

Dear Author, thank you for addressing the round 2 reviews. The manuscript is much improved and very nice now. All the best, Chris